# ROBUST AND SCALABLE SDE LEARNING: A FUNCTIONAL PERSPECTIVE

**Scott Cameron**
Oxford University, InstaDeep Ltd.
United Kingdom
s.cameron@instadeep.com

**Tyron Cameron**
Discovery Insure
South Africa

**Arnu Pretorius**
InstaDeep Ltd.
South Africa

**Stephen Roberts**
Oxford University
United Kingdom

## ABSTRACT

Stochastic differential equations provide a rich class of flexible generative models, capable of describing a wide range of spatio-temporal processes. A host of recent work looks to learn data-representing SDEs, using neural networks and other flexible function approximators. Despite these advances, learning remains computationally expensive due to the sequential nature of SDE integrators. In this work, we propose an importance-sampling estimator for probabilities of observations of SDEs for the purposes of learning. Crucially, the approach we suggest does not rely on such integrators. The proposed method produces lower-variance gradient estimates compared to algorithms based on SDE integrators and has the added advantage of being embarrassingly parallelizable. This facilitates the effective use of large-scale parallel hardware for massive decreases in computation time.

## 1 INTRODUCTION

Stochastic differential equations (SDEs) are a natural extension to ordinary differential equations which allows modelling of noisy and uncertain driving forces. These models are particularly appealing due to their flexibility in expressing highly complex relationships with simple equations, while retaining a high degree of interpretability. Much work has been done over the last century focussing on understanding and modelling with SDEs, particularly in dynamical systems and quantitative finance (Pavliotis, 2014; Malliavin & Thalmaier, 2006). Examples of such work are the Langevin model of stochastic dynamics of particles in a fluid, stochastic turbulence (Kumar et al., 2020), and the Black–Scholes model. More recently, SDE models have gathered interest in machine learning and current work focuses on efficient and scalable methods of inferring SDEs from data.

### 1.1 THE SDE LEARNING PROBLEM

Consider an Itô SDE (Pavliotis, 2014) of the following form

$$\mathrm{d}X = f(X, t)\,\mathrm{d}t + g(X, t)\,\mathrm{d}W, \tag{1}$$

and a set of observations $x_{1:N} = \left\{(t_n, x_n)\right\}_{n=1}^{N}$. In the simplest case, these observations would be directly of a realisation of the process at discrete points in time. In many physical applications, the observations themselves would have some uncertainty associated with them, described by an observation noise model. In more complex models, the SDE would be driving some unobserved parameters describing further complex processes (Li et al., 2020). For example, discrete events may be modeled by an inhomogenious Poisson processes, where the rate parameter is evolving stochastically over time. In any case, the generating process can be summarized by the graphical model in the following diagram and the corresponding learning problem can be described as: "given observations $x_{1:N}$, infer $f$ and $g$".

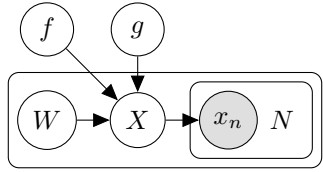

For the most part, the model can be trained using standard techniques, such as variational inference, Markov chain Monte Carlo or optimizing an appropriate loss function. Generally, this requires estimating an expectation over sample-paths, either to estimate the marginal probability of the observations as a function of $f$ and $g$, or to estimate the mismatch of sample-paths with the observations. To estimate these expectations, most current work makes use of integration algorithms. These algorithms generate trajectories of $X$ for a given $f$ and $g$, which can be used to estimate the required expectation. For example, the Euler–Maruyama algorithm (Li et al., 2020; Jia & Benson, 2019; Liu et al., 2019) generates trajectories with the following update rule

$$X_{t+\Delta t} \leftarrow X_t + f(X_t, t)\, \Delta t + g(X_t, t)\, \mathcal{N}\,(0, \Delta t). \tag{2}$$

The expectation of some functional $F$ is estimated using the sample mean of its value over many independent trajectories $X^{(k)}$

$$\hat{F} := \frac{1}{K} \sum_{k=1}^{K} F[X^{(k)}]. \tag{3}$$

Unfortunately, integrators such as Equation (2) introduce a sequential dependence between each point and the next, which inhibits parallelization. Furthermore, it introduces difficulties in probabilistic modelling of the functions $f$ and $g$; in order to properly estimate the expectation, the sample paths should be generated under consistent realisations of $f$ and $g$. Models which implicitly describe $f$ as a Gaussian process would need to sample $f(X_t, t)$ conditioned on all the previously sampled values of $f$ along the trajectory, which may be computationally prohibitive.

The integrator generates trajectories sampled from the prior distribution over paths. In the initial stages of learning, the path-space prior distribution usually has minimal overlap with the path-space posterior and typical trajectories do not pass anywhere near the observations, resulting in high variance gradients. This grows considerably worse as the dimension of the problem and the time between observations increases.

The central idea of this paper is to instead generate trajectories from a distribution much closer to the path-space posterior distribution when estimating these expectations. In the simplest case where the observations are directly of the process $X$, this means sampling trajectories which pass directly through the observations. Alternatively, one can marginalize out the observation noise distribution — or efficiently estimate some expected loss function — by importance sampling

$$\mathbb{E}_X\big[p(x_{1:N}\,|\,X)\big] = \mathbb{E}_{\tilde{x}_{1:N}\sim q}\left[\frac{p(x_{1:N}\,|\,\tilde{x}_{1:N})}{q(\tilde{x}_{1:N})}p_X(\tilde{x}_{1:N})\right], \tag{4}$$

where $q$ is an importance distribution which may depend on the observations themselves. For a large class of observation distributions, such as additive Gaussian noise or multinomial observations, a natural choice for $q$ would be the corresponding conjugate distribution, in which case the importance weight $p/q$ becomes a constant. In complex graphical models, $q$ might be parameterized and optimized variationally, or Gibbs sampling may be used with a collection of simpler conditional distributions. Taking advantage of this, we shall, without loss of generality, concentrate only on estimating the quantity $p_X(x_{1:N}) = \mathbb{E}_X\big[\prod_n \delta\big(x_n - X(t_n)\big)\big]$.

In this work, we introduce a novel algorithm for efficient estimation of probabilities in SDE models for the purposes of learning. Our algorithm has a number of favourable charateristics:

- Our algorithm exhibits lower gradient variance in estimated expectations than algorithms based on SDE solvers, since it enforces that sampled trajectories pass through the observed data, producing accurate results even in hundreds of dimensions.

- Since our algorithm does not rely on an SDE solver, it completely removes all sequential dependence between sampled points. Our algorithm is therefore trivially parallelizable, and can take full advantage of modern parallel hardware such as multi-GPU training.

- Since our estimator can be calculated in a single forward pass of the function $f$, probabilistic approaches such as variational Gaussian processes are viable candidates for the representation of $f$.

## 1.2 Related work

Li et al. (2020) propose a method for calculating gradients of functions of the integrated process with respect to parameters of $f$ and $g$ via a backwards SDE called the adjoint method. This approach requires reconstructing the Wiener process backwards in time when integrating the backward equation and the authors propose a memory efficient algorithm to do so. Their approach uses a constant amount of memory independent of the number of integration steps similar to the adjoint method for NeuralODEs (Chen et al., 2018). Tzen & Raginsky (2019) discuss the computation of gradients by simulation of a forward SDE in the derivatives with respect to parameters of the model using the chain rule of stochastic calculus. This approach does not require back-propagating through the SDE solver. Kidger et al. (2021) propose a method of learning SDEs by treating the SDE as the generator in a generative adversarial network setup, using a second SDE for the discriminator. All of these approaches use SDE integration algorithms to estimate path-space expectations.

Batz et al. (2018) consider a nonparametric approach to SDE learning, using Gaussian processes for the drift and diffusion functions. They initially consider a gradient matching approximation, in which the drift Gaussian process may be fit directly by conventional methods, and thereafter use an expectation-maximization algorithm, and estimate the transition probability for sparse observations using a linear approximation of the process. Yildiz et al. (2018) propose an alternative approach to Gaussian process-based SDE learning using the Euler–Maruyama integrator, approximating $f$ by the predictive mean of a Gaussian process conditioned on a MAP estimate for a set of inducing points. Unfortunately, this approach completely ignores any uncertainty in the posterior over $f$.

It should be noted that each of the above mentioned studies rely on SDE integrators to generate trajectories and therefore face the same difficulties associated with them. Furthermore, almost all of the experiments therein are on system of dimension between 1 and 3, with a small number of experiments of dimension as high as 6. We are unaware of any work which trains neural SDEs in higher dimensions, and we believe this is a testament to the inherent difficulty of learning SDEs in high dimensions.

Much other work focuses on learning specific classes of SDE or ODE models, such as symplectic or Hamiltonian systems (Zhong et al., 2020; Greydanus et al., 2019), graph-based models (Poli et al., 2021; Sanchez-Gonzalez et al., 2019), and controlled differential equations (Morrill et al., 2021). Many of these ideas are independent of the learning algorithm and can potentially make use of our algorithm when applied to SDEs.

## 2  Path-space importance sampling

It is well known that solutions of linear SDEs are Gaussian processes (see Pavliotis, 2014, Section 3.7). For these processes, the joint probability density over some finite collection of points is available in closed form. Furthermore, the posterior distribution of the process conditioned on some set of observations can be sampled from exactly. Unfortunately, this is not the case in general for non-linear SDEs, and one has to resort to approximation methods in order to calculate probabilities. Most commonly, one uses an SDE integrator to estimate expectations; however, other methods, such as Laplace approximations and perturbation theory are sometimes used (Karimi & McAuley, 2016; Dass et al., 2017; Hutzenthaler & Jentzen, 2020).

Alternatively, if we are able to find a second process which follows a similar SDE to the one of interest, it is possible to express expectations and probabilities as importance sampling estimators. Linear SDEs are an attractive candidate for this due to their Gaussianity and ease of sampling.

### 2.1  State-independent diffusion

For now, consider the simpler scenario in which the diffusion coefficient does not depend on the state variable; i.e. $g(x, t) = \sigma(t)$ for some function $\sigma$. The process defined by

$$\mathrm{d}Y = \sigma(t)\,\mathrm{d}W, \tag{5}$$

is Gaussian with mean zero and conditional variance

$$\mathbb{E}_Y\left[\left(Y(t_1) - Y(t_0)\right)^2\right] = \int_{t_0}^{t_1} \sigma(t)\sigma^T(t)\,\mathrm{d}t. \tag{6}$$

Sample paths of $Y$ can be generated efficiently by simulating Brownian motion. The process $Y$ conditioned on a set of observations of the process is a Brownian bridge. If we are able to express quantities of interest as expectations under this process instead of our original SDE, then we can efficiently estimate such quantities via Monte Carlo sampling. As it turns out, expectations and probabilities for an SDE with a general drift term can in fact be expressed as such. This relation is given in the following theorem.

**Theorem 1** *Let $X$ and $Y$ be the stochastic processes generated by the following SDEs*

$$\mathrm{d}X = f(X, t)\,\mathrm{d}t + \sigma(t)\,\mathrm{d}W, \tag{7}$$
$$\mathrm{d}Y = \sigma(t)\,\mathrm{d}W. \tag{8}$$

*Further assume that $\sigma(t)\sigma^T(t)$ is Lebesgue-almost-everywhere invertible and $f$ is sufficiently well-behaved such that $X(t)$ has a unique probability density for all $t$.*

*The probability density of observations $\left\{(t_n, x_n)\right\}_{n=1}^{N}$, under the process $X$ is given by the conditional expectation*

$$p_X(x_{1:N}) = p_Y(x_{1:N})\mathbb{E}_Y\left[e^{S[Y]}\,\middle|\,\left\{Y(t_n) = x_n\right\}_{n=1}^{N}\right], \tag{9}$$

*where*

$$S[Y] = \int f^T(Y_t, t)\big(\sigma(t)\sigma^T(t)\big)^{-1}\,\mathrm{d}Y_t - \frac{1}{2}\int f^T(Y_t, t)\big(\sigma(t)\sigma^T(t)\big)^{-1}f(Y_t, t)\,\mathrm{d}t. \tag{10}$$

This result follows from Girsanov's theorem (Girsanov, 1960; Malliavin & Thalmaier, 2006, Section 1.5) and the definition of conditional expectation. Intuitively speaking, the first term in Equation 10 encourages $f$ to line up with typical paths which pass through the observations, while the second term regularizes the magnitude of $f$.

Theorem 1 allows us to develop an importance sampling algorithm for estimating probabilities under the process $X$ by simulating the process $Y$, where the importance weights are given by $\omega_i = e^{S[Y^{(i)}]}$. In this approach, one can generate entire sample paths of $Y$ before calculating $f(Y, t)$, allowing the forward pass of $f$ to be performed in parallel. This approach is described in Algorithm 1.

---

**Algorithm 1** Path Integral Importance Sampling

---

1: **for** $i = 1, \ldots, K$ **do**
2:      Sample $(Y_t)_{t=t_1}^{t_N}$, s.t. for each $n$, $Y_{t_n} = x_n$                  $\triangleright$ $Y$ is a Brownian bridge
3:      $f_t \leftarrow f(Y_t, t)$ for each $t$
4:      $\alpha \leftarrow \sum_t f_t^T \sigma^{-2}(t)\,(Y_{t+\Delta t} - Y_t)$
5:      $\beta \leftarrow \sum_t f_t^T \sigma^{-2}(t) f_t\,\Delta t$
6:      $S_i \leftarrow \alpha - \frac{1}{2}\beta$
7: **end for**
8: **return** $p_Y(x_{1:N}) \frac{1}{K}\sum_i \exp(S_i)$

---

An important property of this algorithm is that the sampled paths directly pass through observations. This means that the probability estimates directly relate the drift function $f$ to plausible trajectories instead of allowing the particles to wander around randomly. See Figure 1 for an illustration. In the initial stages of training, simply integrating the SDE with drift $f$ will typically not lead to trajectories which pass near the data, especially in high dimensions. This can have a significant impact on the variance of the gradients and can inhibit the model's ability to learn. Training on paths that pass through the data can greatly mitigate this high variance. As with other algorithms that generate SDE sample paths, the discretization of the integrals introduces a small bias. However, this bias vanishes in the $\Delta t \to 0$ limit.

### 2.1.1    A NOTE ON PARALLELIZABILITY

With the exception of the sampling step in Line 2 of Algorithm 1, each of the other operations, including the sums, can be performed in parallel over feature and batch dimensions, as well as over

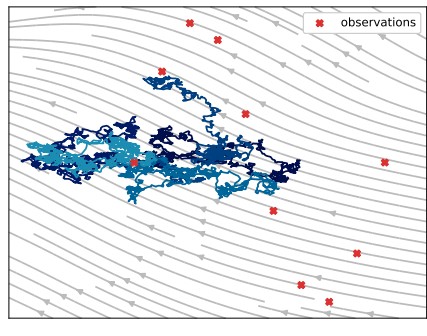
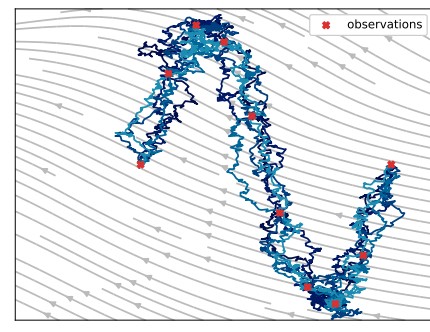

(a) SDE integration                     (b) Brownian bridge importance samples

Figure 1: Independently sampled paths. a shows paths sampled from the SDE using an integrator, while b shows samples from a Brownian bridge which pass exactly through the observations. In both cases the drift function is given by a neural network which is randomly initialized with the same seed. The drift vector field is shown in grey.

the $t$ and $i$ indices. Sampling a Brownian bridge, or any conditional Gauss–Markov process, can be performed by first sampling the unconditional process, and then linearly interpolating with the observations. The linear interpolation can be performed independently, and therefore in parallel, over the time dimension. The only operation that is not completely parallelizable is the sampling of the unconditional base process — in this case standard Brownian motion. Brownian paths can be sampled by cumulative summation of independent normal random numbers. However, this cumulative sum is extremely fast, and can be performed independently between each consecutive pair of observations. In our tests, sampling whole Brownian trajectories was about 5 orders of magnitude faster than the forward pass of the neural network $f$, and so does not create any noticeable performance impact.

Once the Brownian paths are sampled, the rest of Algorithm 1 may be implemented in parallel without difficulty. Contrast this to standard integrator-base methods; even if the Brownian paths are sampled upfront, the integrator must still perform each of the forward passes of $f$ in sequence.

See Appendix B for more details on sampling Gaussian bridges.

## 2.2 STATE-DEPENDENT DIFFUSION

The assumption that $g$ is independent of the state is not in fact required for the validity of Theorem 1; the reason this assumption is required is to ensure that the process $Y$ is Gaussian, so as to enable easy conditional sampling and calculation of $p_Y(x_{1:N})$. Unfortunately, one cannot simply find an analogue of Theorem 1 to calculate expectations of a general process as expectations with respect to a constant diffusion process.

To address the question of state-dependent diffusion, it is enlightening to consider a simple example of such a process. Perhaps the most common example is geometric Brownian motion

$$\mathrm{d}X = \mu X \, \mathrm{d}t + \sigma X \, \mathrm{d}W. \tag{11}$$

The typical way in which one would solve this is to introduce a transformation $Z = \log(X)$, and apply Itô's lemma (see Malliavin & Thalmaier, 2006, Section 1.5; Pavliotis, 2014, Section 3.5) to obtain the SDE

$$\mathrm{d}Z = \left(\mu - \tfrac{1}{2}\sigma^2\right) \mathrm{d}t + \sigma \, \mathrm{d}W. \tag{12}$$

Similarly, one might address a problem of a general state-dependent diffusion coefficient by considering a transformation to a space in which the transformed equation has constant diffusion. Let $T(\cdot, t)$ be a map which is invertible for all $t$. That is, there exists a function $T^{-1}$ such that if $y = T(x, t)$, then $x = T^{-1}(y, t)$. Transforming a process with constant diffusion $\sigma$, by the map $T^{-1}$ gives a process with diffusion coefficient

$$g_{i,j}(x,t) = \frac{\partial T_i^{-1}}{\partial y_k}\sigma_{k,j} = \left(\frac{\partial T(x,t)}{\partial x}\right)^{-1}_{i,k}\sigma_{k,j}, \tag{13}$$

where repeated indices imply summation. With such a transformation we can represent any diffusion matrix that can be written as a Jacobian. While the restrictions on the diffusion matrix may be seen as an inherent limitation of a path-space importance sampling approach, it may be argued that constant or simple diffusion functions would cover the majority of cases of interest, and almost all real-world problems require some form of data pre-processing and, potentially, link functions.

This transformation allows us to infer the SDE in a transformed space, with constant diffusion, and, if desired, transform the equation back to the original space using Itô's lemma. This approach is described in Algorithm 2, where we have used the symbol $\tilde{f}$ to represent the drift in the transformed space as to avoid ambiguity with the drift in the original space. One important point to note is that one should be careful to allow gradients to propagate through the sampling step on Line 2 in Algorithm 2, so as to allow learning of the function $T$.

---

**Algorithm 2** Transformed-State Path Integral Importance Sampling

---

1: **for** $i = 1, \ldots, K$ **do**
2:     Sample $(Y_t)_{t=t_1}^{t_N}$, s.t. for each $n$, $Y_{t_n} = T(x_n, t_n)$                  $\triangleright$ $Y$ is a Brownian bridge
3:     $\tilde{f}_t \leftarrow \tilde{f}(Y_t)$ for each $t$
4:     $\alpha \leftarrow \sum_t \tilde{f}_t^T \sigma^{-2} \left(Y_{t+\Delta t} - Y_t\right)$
5:     $\beta \leftarrow \sum_t \tilde{f}_t^T \sigma^{-2} \tilde{f}_t \, \Delta t$
6:     $S_i \leftarrow \alpha - \frac{1}{2}\beta$
7: **end for**
8: **return** $\prod_n \det\left(\frac{\partial T(x_n, t_n)}{\partial x_n}\right) p_Y\left(T(x_{1:N}, t_{1:N})\right) \frac{1}{K} \sum_i \exp(S_i)$

---

One would typically apply this method to infer $\tilde{f}$ rather than the original $f$ for performance reasons, and then only reconstruct the SDE in the original space if required.

### 2.2.1 RECONSTRUCTION OF THE SDE IN THE UNTRANSFORMED SPACE

Generation of new sample paths can be performed easily in the transformed space and then simply transforming back by applying $T^{-1}$. However, sometimes one needs more information for the purposes of analysing the learned equation. In this case the SDE can be transformed back using Itô's lemma. The drift and diffusion coefficients are

$$f_i(x, t) = \frac{\partial T_i^{-1}}{\partial t} + \frac{\partial T_i^{-1}}{\partial y_k} \tilde{f}_k(y, t) \; + \; \frac{1}{2}\sigma_{j,k} \frac{\partial^2 T_i^{-1}}{\partial y_j \partial y_l} \sigma_{l,k}, \tag{14}$$

$$g_{i,j}(x, t) = \frac{\partial T_i^{-1}}{\partial y_k} \sigma_{k,j}. \tag{15}$$

All expression are evaluated at $y = T(x, t)$. When $T$ is parameterized by a neural network with ReLU non-linearities, the second derivative term in Equation (14) is almost-everywhere zero. The remaining terms are Jacobian vector products, and so can be performed efficiently even in high dimensions.[1]

## 3 EXPERIMENTS

In this section, we provide some experiments to illustrate and verify the performance and capabilities of our proposed algorithm. Further details on the experiments are given in Appendix C.

### 3.1 THE LORENZ SYSTEM

To compare the effect of our algorithm on learning, we follow Li et al. (2020) and train a neural network based SDE model on the Lorenz system. In this model the drift function is given by a

---

[1]However, this may require forward-mode instead of backward-mode differentiation.

multi-layer perceptron, and the diffusion term is a simple scalar. We generated 16 independent paths, consisting of 200 observations each, uniformly sampled over a time interval of $[0, 10]$. We trained the model on this data by two different methods. In the first method, we minimize the mean-squared error, estimated by integrating out the SDE, and calculating gradients using the adjoint method (Li et al., 2020).[2] In the second method, we directly optimize the log-probability calculated using Algorithm 1. In both cases we used precisely the same optimization parameters, the same time-step size $\Delta t$, and the same number of sampled trajectories to calculate averages. Both models used the same seed for the neural network initialization, and the same number of gradient steps were performed in both cases. These models were trained on an Nvidia RTX 3070, with 8Gb of memory.

To assess the performance of these approaches, we record two metrics. The first is the mean-square difference between the learned drift vector field, and the Lorenz vector field, normalized by the mean-square magnitude of the Lorenz vector field, measured at the observations. Written mathematically:

$$\text{err} = \frac{\sum_n \big(f_\theta(x_n) - f_{\text{Lor}}(x_n)\big)^2}{\sum_n \big(f_{\text{Lor}}(x_n)\big)^2} \tag{16}$$

This metric is shown in Figure 2b. The second metric we record is the mean-squared error of sampled trajectories compared to the observations; this is shown in Figure 2a. However, we note that this is not a fair performance metric since the adjoint-based method is directly optimizing for it. On the x-axis we show the training time in minutes.

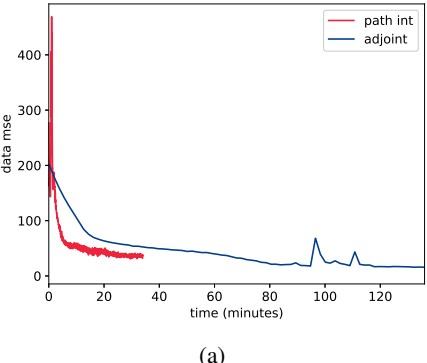
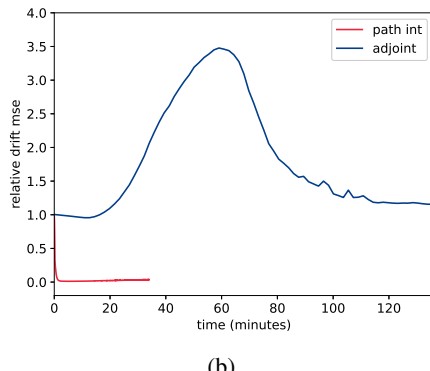

(a)                                                (b)

Figure 2: Learning curves of a neural SDE on the Lorenz system. b shows the normalized mean-squared deviation of the learned drift from the ground truth. a shows the mean-squared error of sample paths from the observations.

In Figure 2 we have limited the range of the x-axis in order for the lines to be more clearly visible. The total training time for the adjoint-based MSE optimization was a factor 52 longer than the time taken optimizing the probability estimates. We note that, while our approach was significantly faster, it did use approximately four times as much memory as the adjoint-based method.[3]

The poor performance of the integrator-based learning algorithm in Figure 2b — despite its low mean-squared-error — suggests that directly optimizing for the discrepancy of typical SDE sample-paths compared to observations is not a robust approach to SDE learning.

## 3.2 GRADIENT VARIANCE

We applied both our algorithm and an SDE integrator to a neural network-based SDE model on a real-world dataset and recorded the gradients, sampled over many independent trajectories. We thereafter computed the variance of the gradients individually per parameter, for various time lengths between observations. We used a discretization step size of 0.01. These results are shown in Figure 3.

---

[2]We also attempted training a model without the adjoint method and instead directly backpropagating through the SDE integrator. However, a single forward pass required more memory than available on the GPU.

[3]The adjoint algorithm was specifically designed to have constant memory complexity in the number of integration steps. Our algorithm does not exhibit this feature.

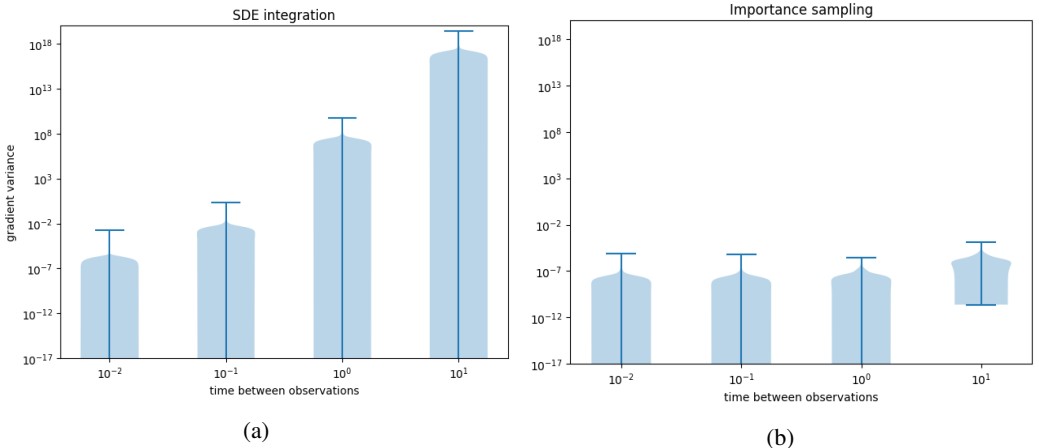

Figure 3: Gradient variances as a function of the time between observations. a shows gradient variances of the observation mean-squared-error computed with an SDE integrator. b shows the gradient variance of the log-probability of observations computed with Algorithm 1.

In each case we used the same neural network, initialized with a fixed seed, and the exact same values for all parameters. We used the same loss functions as in Section 3.1. The data set we used was the Hungarian chickenpox cases dataset from the UCI machine learning repository;[4] it has 20 features per observation. We scaled the features by their standard deviations before performing the experiment to improve numerical stability.

Our findings are as follows:

- For very short times, we found that both algorithms produced comparable gradient variances: typically less than $10^{-7}$.

- For longer time spans (3 orders of magnitude larger than the discretization step size), the gradient variance of our algorithm increased by a few orders of magnitude; the maximum we observed was on the order of $10^{-4}$. However, the SDE integrator approach resulted in huge increases in variances, with some as large as $10^{18}$.

- Between the above two extremes, we found that the gradient variance of our approach did increase with the time between observations, but stayed below $10^{-5}$, while the gradient variance from the SDE integrator exhibited very poor scaling with variances of up to $10^8$ for times between observations of 1.0.

Considering that both approaches produced comparable variances on very small timescales, we feel that this experiment represents a fair comparison.

### 3.3 UNCERTAINTY QUANTIFICATION WITH VARIATIONAL GAUSSIAN PROCESSES

We attempt to test the viability of using our method for an SDE model with a GP prior over the drift function and fitting the posterior variationally. To assess this, we used data trajectories from a van der Pol oscillator. For the GP prior, we used a zero mean function and a radial basis function kernel, with independent GP priors over each output dimension of the drift vector field. These choices are not necessarily optimal; however, the purpose of this experiment is merely to illustrate the feasibility of using variational GP to learn the vector field of an SDE model.

We used a grid-based inducing point approach and the kernel interpolation method of Wilson & Nickisch (2015). The combination of these techniques allows one to take advantage of the Kronecker product structure of the covariance matrices and the Toeplitz structure of each factor for accelerated sampling. The model was trained variationally, sampling the variational posterior jointly along entire trajectories.

---

[4]This dataset can be found at https://archive.ics.uci.edu/ml/datasets/Hungarian+Chickenpox+Cases

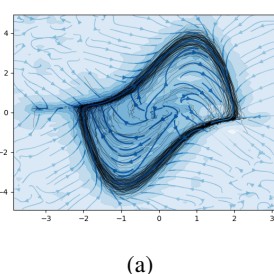
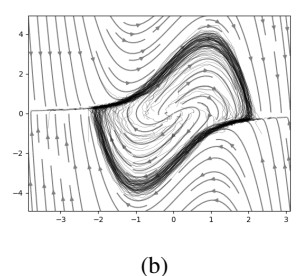
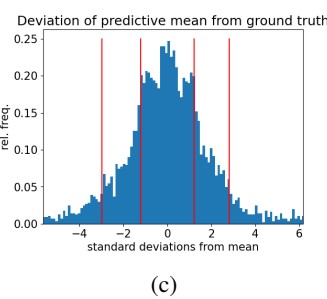

(a)                                    (b)                                    (c)

Figure 4: GP posterior compared to ground truth vector field. a shows the GP vector field coloured by uncertainty. The arrows show the integral curves of the posterior mean and the coloured are scaled linearly with the log predictive variance. b shows the integral curves of the ground truth van der Pol oscillator. c shows the deviation of GP posterior from the ground truth vector field in standard deviations. The red lines show the 10%, 25%, 75%, and 90% quantiles.

The approximate GP posterior is shown in Figure 4a compared to the ground truth vector field of the van der Pol oscillator in Figure 4b. Comparing the GP to the van der Pol vector field, we found that the GP model was fairly accurate in areas with many data points, while reverting to the prior far away from observations. To assess the quality of the uncertainty quantification, we further calculated the discrepancy between the variational posterior and the ground truth vector field. Over a dense grid, covering an enlarged area around the observations, we calculated the discrepancy as

$$\delta = \frac{f_{\text{vdp}}(x) - \mu(x)}{\sigma(x)}. \tag{17}$$

A histogram is shown in Figure 4c.

Under Gaussianity assumptions, one expects to see the quantiles line up well with some associated number of standard deviations. However, this will only be the case near to the observations. In regions where the GP reverts to the prior, which has a stationary kernel, the mismatch between the GP and the polynomial growth of the van der Pol oscillator results in heavy tails. We expect that a better choice of kernel, which accounts for this polynomial behaviour, to show residuals with lighter tails. We found that the distribution of these $\delta$s was fatter-tailed than a Gaussian distribution; the middle 50% (25% - 75% quantiles) slightly passed one standard deviation on either side, as opposed to the 68% of a standard Gaussian distribution. The middle 80% (10% - 90% quantiles) were within 3 standard deviations. While the posterior uncertainty was somewhat overconfident, it was not wholly unreasonable.

We further note that, while the complexities involved in variational GP fitting can result in longer training times, we still found that our method applied to a GP was significantly faster than using an SDE integrator with a neural network.

## 4  DISCUSSION

In this work we introduced an algorithm to efficiently estimate probabilities of observations under the assumption of an SDE model for the purposes of learning. The proposed approach produces estimates which are accurate, even in high dimensions, and can fully utilize parallel hardware for significant speedups.

The proposed algorithm is capable of providing robust probability estimates which can be used in models both with or without an explicit observation noise component. This is in contrast to SDE integrator-based learning approaches, which must incorporate and estimate an explicit error term due to their inability to generate paths which are guaranteed to pass through the observations.

Our algorithm produces more stable estimates than alternative integrator-based approaches, particularly when the observations are sparsely distributed over long time intervals and in high dimensions. This in turn results in more stable and robust learning. With the rise in applicability of large-scale (neural network) SDE modeling in science, simulation, and domains such as finance, we see our approach as an important contribution to reliable scaling of these methods.

ACKNOWLEDGMENTS

The authors would like to thank InstaDeep Ltd. for their financial support of this work.

REPRODUCIBILITY

Efforts have been made to describe all parameters and details of the algorithms used in this paper, either in the content or in the appendix. The code included in the supplementary material includes a Dockerfile and instructions for running some of the experiments. Where relevant, fixed seeds for random number generators can be found in the code during initialization.

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

## A    Theorem 1

We have that $X$ and $Y$ are processes obeying the SDEs

$$dX = f(X)\,dt + g(X)\,dW, \tag{18}$$
$$dY = g(Y)\,dW, \tag{19}$$

dropping the time argument for notational simplicity. Expectations with respect to SDEs can be expressed as path integrals. In the Itô convention, the path integral representation of an SDE has the following form

$$\mathbb{E}_X\big[F[X]\big] = \lim_{N\to\infty} \int F[x] \prod_{n=1}^{N} \frac{dx_n}{\sqrt{2\pi\Delta t_n}\,g(x_{n-1})} \exp\left\{ -\frac{1}{2} \sum_{n=1}^{N} \left( \frac{\Delta x_n - f(x_{n-1})\Delta t_n}{g(x_{n-1})\sqrt{\Delta t_n}} \right)^2 \right\} \tag{20}$$

$$=: \int \mathcal{D}x F[x] J \exp\left\{ -\frac{1}{2} \int \mathcal{L}(x(t), \dot{x}(t))\,dt \right\}, \tag{21}$$

where $J$ is a Jacobian factor which depends on $g$ but not $f$ and the Lagrangian is given by

$$\mathcal{L}(x, \dot{x}) = \left( \frac{\dot{x} - f(x)}{g(x)} \right)^2.$$ (22)

The derivative of $x$ here may be thought of distributionally; however, the expression for the measure $\mathcal{D}x \exp\{-\frac{1}{2} \int \mathcal{L}\}$ need only be defined on the Cameron–Martin subspace[5]; guaranteeing its unique extension to the whole Wiener space. In contrast to this, the expression inside the expectation $F[x]$ should be defined almost-everywhere, and hence care should be taken when $\dot{x}$ appears in an expectation.

We may expand the quadratic to separate the Lagrangian into a drift-free term plus terms dependent on the drift

$$\mathcal{L} = \left( \frac{\dot{x}}{g(x)} \right)^2$$ (23)

$$- 2f^T(x)g^{-2}(x)\dot{x}$$ (24)

$$+ f^T(x)g^{-2}(x)f(x).$$ (25)

The first term, on line 23, is the Lagrangian for the process $Y$, which has no drift term. Shuffling the rest of the terms out of the measure and into the expectation, we have

$$\mathbb{E}_X\big[F[X]\big] = \int \mathcal{D}x F[x] e^{S[x]} \exp\left\{ -\frac{1}{2} \int \left( \frac{\dot{x}}{g(x)} \right)^2 \mathrm{d}t \right\},$$ (26)

where

$$S[X] = \int f^T(X_t)g^{-2}(X_t)\dot{X}_t \, \mathrm{d}t$$ (27)

$$- \frac{1}{2} \int f^T(X_t)g^{-2}(X_t)f(X_t) \, \mathrm{d}t$$ (28)

$$= \int f^T(X_t)g^{-2}(X_t) \, \mathrm{d}X_t$$ (29)

$$- \frac{1}{2} \int f^T(X_t)g^{-2}(X_t)f(X_t) \, \mathrm{d}t.$$ (30)

Note that on the third line we have used the distributional definition of $\dot{X}$; i.e. $\dot{X}_t \, \mathrm{d}t = \mathrm{d}X_t$.

Lastly, using the definition of conditional expectation, we have that

$$\mathbb{E}_A\Big[F[A]\, p(B = b|A)\Big] = \mathbb{E}_A\Big[p(B = b|A)\Big]\mathbb{E}_A\Big[F[A]\,\Big|\, B = b\Big].$$ (31)

By trivial substitution, we have

$$p_X(x_{1:N}) = p_Y(x_{1:N})\mathbb{E}_Y\left[e^{S[Y]}\,\Big|\, \{Y(t_n) = x_n\}_{n=1}^N\right].$$ (32)

## B    SAMPLING BRIDGES

Let $Y$ be a Gauss–Markov process with kernel $k$. Since the process is Markov, we need only sample the process conditioned on two end-points. Let $y = (y_0, y_T)$ be the known values of the process at times $\tau = (t_0, T)$. Let $\tilde{Y}$ be the process defined by

$$\tilde{Y}_t := Y_t + K(t, \tau)K(\tau, \tau)^{-1}\big(y - Y_\tau\big), \quad \text{for} \quad t_0 \le t \le T,$$ (33)

where $K$ is the matrix whose entries are $k(t_i, t_j)$. This distribution of $\tilde{Y}$ is the same as the distribution of $Y$ conditioned on the values $y$ at times $\tau$. Note that once $Y_t$ has been sampled — which may

---

[5]The Hilbert space of functions whose derivatives are square integrable. The Wiener space is the space of continuous functions. This is not a Hilbert space, instead it is a Banach space with the supremum norm. The measure of the Cameron–Martin space is in fact zero; the sample paths of the SDE are almost-surely nowhere differentiable.

require a cumulative sum — this transformation can be performed independently, and hence in parallel, for each $t$. For each consecutive pair of observations, the bridge process can be sampled independently.

In the simplest case, where $Y$ is a Brownian motion starting from $y_0$ at time $t_0 = 0$, this reduces to

$$\tilde{Y}_t = Y_t + \frac{k(t,T)}{k(T,T)}(y_T - Y_T) \tag{34}$$

$$= Y_t + \frac{t}{T}(y_T - Y_T). \tag{35}$$

And similarly for any other processes which can be sampled with a fixed starting point.

In the more general case we can invert the two by two matrix to get an explicit formula for $\tilde{Y}_t$

$$\tilde{Y}_t = Y_t + \frac{k(t,t_0)k(T,T) - k(t,T)k(t_0,T)}{k(t_0,t_0)k(T,T) - k(t_0,T)^2}(y_0 - Y_0) \tag{36}$$

$$+ \frac{k(t,T)k(t_0,t_0) - k(t,t_0)k(t_0,T)}{k(t_0,t_0)k(T,T) - k(t_0,T)^2}(y_T - Y_T). \tag{37}$$

This may be extended straightforwardly to other cases, such as conditioning on the values of the integral, or derivative of the process etc.

## C  EXPERIMENTS

### C.1  PROBABILITY ESTIMATES

As a sanity check, we compare the log-probability estimates of our algorithm to the actual log-probabilities of an Ornstein–Uhlenbeck process. The probability density for this process is available in closed form for any number of observations. We estimate the probability of a set of data points evenly spaced over a time interval of $[0, 10]$, taken from a sine function $x_n = \sin(\pi/10 \times t_n)$. For these estimates we used $1000$ time steps to estimate the integral $S$, and averaged over $100$ trajectories, independently sampled from a Brownian bridge passing through the observations. We repeated this process $1000$ times and report the mean and standard deviation of these estimates. Figure 5 shows these estimates compared to the analytic probability given by the Ornstein–Uhlenbeck process for various numbers of observations as a function of the dimension of the problem.

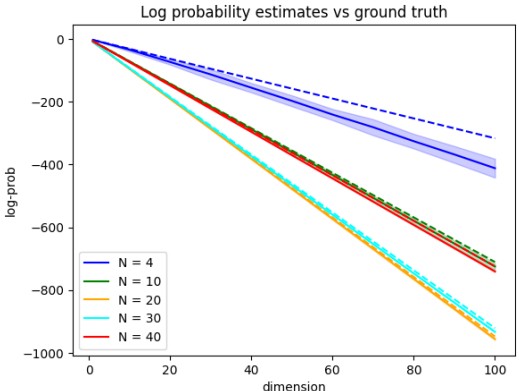

Figure 5: Log-probability estimates for an Ornstein–Uhlenbeck process. The solid lines indicate the mean value of the estimator, and the shaded regions indicate three standard deviations from the mean. The dashed lines are the analytic log-probability values.

An interesting observation is that the variance of the estimator seems to decrease with the number of data points. This is atypical of probability estimators, since the magnitude of the log-probability increases linearly with the number of observations (not to be confused with the number of samples used to estimate the log-probability). In almost all cases, with the exception of 4 observations in more than $40$ dimensions, our algorithm gives accurate log-probability estimates with small standard

deviations. The error of this exception is approximately 30%, but is significantly reduced by adding more observations, or using more samples. We believe this error is acceptable considering the small number of observations and the large scale. For 10 observations in 100 dimensions the standard deviation of our estimator is only 0.4% of the magnitude of the log-probability and the error of the mean estimate is 2% of the analytic log-probability. This small bias can be further reduced by using more samples or a smaller $\Delta t$. A single estimate in the 100 dimensional case took less than half a second to compute on a laptop CPU, and is therefore feasible to use in the inner loop of a learning algorithm.

## C.2 LORENZ

### C.2.1 DATA SET

We generated data by simulating a Lorenz system with the exact same parameters as Li et al. (2020). We generated 16 independent trajectories, with initial values sampled from a standard normal distribution, and discretely sampled the values of the process over 200 equally spaced time points ranging from 0 to 10. Unlike Li et al. (2020), we did not add any observation noise to the data.

We deliberately used a sparser data set than Li et al. (2020), due to personal experience of the difficulty of fitting SDE models on sparse observations.

### C.2.2 MODEL

The model we used to fit to the observations was

$$\mathrm{d}X_t = f_\theta(X_t)\,\mathrm{d}t + \sigma\,\mathrm{d}W, \tag{38}$$

where $\sigma$ is a scalar and $f_\theta$ is a fully-connected neural network with ReLU nonlinearities and parameters $\theta$. We used layer sizes of $3 \to 32 \to 256 \to 32 \to 3$.

### C.2.3 TRAINING PARAMETERS

We used precisely the same parameters for each model, the only difference in the training procedures being the loss functions used. In the first case, the loss function per trajectory was

$$L_{\mathrm{mse}}(\theta; x_{1:N}) = \frac{1}{KN} \sum_{k,n} \big(X_{t_n}^{(k)} - x_n\big)^2, \tag{39}$$

where $N = 200$ is the number of observations per trajectory, and $K$ is the number of sampled paths used to estimate the MSE, and $X^{(k)}$ is sampled using an SDE integrator starting at $X_{t_1}^{(k)} = x_1$. We then averaged this loss function over the 16 independent observed trajectories.

In the second case, the loss function per trajectory was

$$L_{\mathrm{PI}}(\theta; x_{1:N}) = -\frac{1}{N} \log \hat{p}_X(x_{1:N} \mid \theta), \tag{40}$$

where $\hat{p}_X(x_{1:N} \mid \theta)$ is calculated using Algorithm 1 with $K$ trajectory samples. Again this loss was averaged over the 16 independent observed trajectories. In both cases we used $K = 64$ and a time step size of $\Delta t = 10^{-2}$.

For each model we used the Adam optimizer with a learning rate of $10^{-3}$ and ran the optimization algorithm for $10^4$ iterations. The model using our importance sampling approach completed the full number of iterations in 34 minutes, while the integrator-based approach took 29 hours and 46 minutes to complete the same number of iterations.

Our approach consistently maintained 100% GPU utilization, except during MSE evaluation which used an integrator, while the integrator based approach was unable to surpass 25% GPU utilization in this experiment.

## C.3 VARIATIONAL GAUSSIAN PROCESSES

Our approach allows more straightforward implementation of SDE models using a Gaussian process prior for the drift function. Some previous approaches often resort to approximations such as

using only the predictive mean or MAP estimates (Yildiz et al., 2018). For dense observations, directly discretizing the SDE and fitting the GP to the observation deltas may be viable. For sparse observations this is much more difficult. Our proposed method makes variational GP methods slightly simpler, since the values of $f(Y_t)$ can be sampled jointly along the path. However, it is important to note that various complexities still arise such as the difficulty of efficiently sampling high-dimensional Gaussian distributions.

We used the kernel interpolation method of Wilson & Nickisch (2015) which stores a grid of inducing locations per dimension. Storing inducing locations per dimension for a Kronecker product kernel effectively gives a variational distribution over $m^d$ inducing points with only $d \times m$ memory requirements, where $m$ is the number of inducing points per dimension and $d$ is the dimension. When these inducing locations are evenly spaces, the covariance factors have a Toeplitz structure which allows sampling in linearithmic time complexity.

To fit the model we used gradient ascent on the ELBO which is calculated as follows:

$$\text{ELBO} = \mathbb{E}_{f \sim q} \big[ \log p(x|f) \big] - \text{KL} \big[ q(f) | p(f) \big]. \tag{41}$$

The KL divergence is between two Gaussian distributions calculated at the inducing locations in the usual way. For the likelihood term, we generated trajectories (as in Algorithm 1), followed by jointly sampling the variational GP posterior along these trajectories and proceed with the rest of the computation in Algorithm 1 using these samples. Our optimization objective is calculated

$$L = -\frac{1}{K} \sum_K \log \hat{p}(x_{1:N} \,|\, f^{(k)}) + \text{KL} \big[ q(f) | p(f) \big], \tag{42}$$

where each $f^{(k)}$ is an independent sample over the whole path of $Y$. In other words, $f^{(k)}(Y_{t_1})$ and $f^{(k)}(Y_{t_2})$ are correlated, but $f^{(k)}(Y_{t_1})$ and $f^{(j)}(Y_{t_2})$ are not when $k \neq j$.

In our experiment we used 32 evenly spaced inducing points per dimension, and jointly sampled the variational posterior over 64 trajectories of 1024 time steps. The model was trained with the Adam optimizer with an initial learning rate of 0.1, and an exponential decay with factor 0.999 for $10^5$ training steps. However, the square deviation of the predictive mean from the ground truth converged well before $10^4$ iterations. Training for $10^4$ iterations took 34 minutes, $10^5$ iterations took about 5 and a half hours.

