# OpenReview forum: "Robust and Scalable SDE Learning: A Functional Perspective"
_ICLR.cc/2022/Conference — ICLR 2022 Poster_

### Official Review · Reviewer_p6RL · 2021-11-02

**Correctness:** 3
**Technical Novelty And Significance:** 3
**Empirical Novelty And Significance:** 2
**Recommendation:** 6
**Confidence:** 2

**Main Review:**

- **Clarity.** The paper is mostly written clearly and succinctly. However, it could be more self-contained, possibly with additional background provided in the supplementary materials.
- **Reproducibility.** Low-level details necessary to reproduce the results from this paper are not all included. However, source code is provided with the supplementary material, which appears to be well-structured and clean.
- **Novelty.** The main technical novelty comes from the application of Theorem 1, as summarized in Algorithm 1. In particular, the authors propose to estimate the original SDE of interest with an linear SDE, which is an attractive alternative since it is easier to estimate their probabilities and expectations with importance sampling, and has the added benefit that the sampled paths pass through the observations. This appears to provide significant improvement, as evidenced by the empirical results.

## Concerns

Overall, I find the empirical evaluations feel a bit rushed.

- Experiment described in Section 3.1: are you making use of parallelism in this experiment? If so, the details of this need to be described. If not, can you pinpoint and provide some explanation for the factor of 52 speed-up? From what I can tell, even without any parallelism, the method is able to provide speed-ups of this magnitude? Furthermore, the plots might be clearer visualized in log-y scale. Also, instead of truncating the training time along the horizontal axis, you might consider instead recording the amount of time required to attain a particular MSE value. As it is currently, the differing lengths of the curves make it slightly messy. Finally, the only baseline that is compared against is the adjoint method of Li et al. 2020. I would be interested in comparing the proposed method against some of the related methods highlighted in the Related Works section as baselines.
- Experiment described in Section 3.3: I find the normality test carried out starting on line 271 to be lacking in precision. Firstly it is entirely qualitative and therefore somewhat subjective. There are simple statistical tools available to carry out normality tests. Line 280 concerning training speeds: the speed-up is mentioned in passing. Though not surprising, it would be good to see this quantified more rigorously with a table or figure, rather than as an offhand remark.

## Miscellaneous Issues

- The statement of contribution at the end of page 2 contains four bullet points. Only the first bullet point is the actual contribution. The remaining three are merely descriptions of the first and only contribution.
- Line 14: "which" → "that"
- Line 46: "Gaussian process, [...]" - extraneous comma
- Line 146: "which" → "that"
- Line 183: "which" → "that"
- Line 222: "The second metric we record the [...]" - missing "is"
- Figure 2: If I understand correctly, the two sub-figures shown here are of the same vector field and observations. Is that right? This would be made much more clear by having the subplots share the same vertical and horizontal axes.

**Summary Of The Paper:**

This paper proposes an importance-sampling method for estimating the likelihood of observations in stochastic differential equation (SDE) models. Unlike classical integrator-based methods, the proposed approach imposes a structure that forces sampled trajectories to pass through observations thereby improving accuracy, especially in high dimensions. Furthermore, it removes sequential dependencies between sampled points thus allowing for speed-ups through parallelism.

**Summary Of The Review:**

Given the numerous advantages of the proposed method over existing works, I am inclined to recommend acceptance of this paper. However, given the questions I raised concerning the empirical evaluations, I am reluctant to recommend a clear acceptance.

---

> ### Author Response · Authors · 2021-11-22
> **Reply to Reviewer p6RL**
>
> Thank you for your feedback. We have attempted to address most concerns and have updated our paper based on them.
>
> On a more specific point; we have attempted various combinations of log axes and scaling of the graphs in the Lorenz experiment. However, we unfortunately did not find that they improved the clarity, but made interpreting the results more difficult. On the other hand, we have adjusted the axes of the plots in figure 2 as suggested.
>
> Regarding the GP experiment, it is true that our normality test is a very naive one. In fact, the distribution of the errors is clearly not Gaussian at all. The residuals will only be Gaussian if an appropriate prior is used which takes into account the polynomial nature of the vector field. However, the quality of the GP model and its ability to fit this system is orthogonal to our contribution. Our real contribution in this experiment is to illustrate that full variational GP models are feasible with our algorithm, in which case we believe that a qualitative assessment of the model would be acceptable. We see that this point was not clearly represented, and we have updated the section in the paper accordingly.

---

> > ### Comment · Reviewer_p6RL · 2021-11-29
> > **Acknowledgement**
> >
> > Thanks for your response. After following the discussions concerning the points raised by the other authors, I have decided to stand by my original overall score.

---

### Official Review · Reviewer_k19Y · 2021-11-02

**Correctness:** 2
**Technical Novelty And Significance:** 3
**Empirical Novelty And Significance:** 2
**Recommendation:** 5
**Confidence:** 2

**Main Review:**

+ The paper indeed addresses a timely topic. Given that continuous-time dynamical system literature is currently dominated by neural ordinary differentia equations, contributions like this would pave the way for more stochastic differential equation publications.
+ The results seem pretty impressive. However, the experiments are done only on very few dimensional systems. More experiments on bigger systems would help us identify trade-offs and potential issues with the methodology.
+ The paper is written clearly and the very obvious motivation is communicated very well. A few notes to further improve the readability:
    - What does the sentence in line 48 mean? More specifically, what prior/posterior distributions do you refer to?
    - More details on the GP-SDE model in Sec 3.3 would be nice. Also, the experiment seems a bit repetitive.
    - Alg 1 should be written more verbosely, e.g., the set of $t$'s (in line 3) and $\Delta t$ can be explicitly given.

These being said, I'm not able to verify Th 1. Despite checking the references and more resources, I could not immediately see (9-10). If the authors can provide a proof (even if it is straightforward) or other reviewers can comment on it, I would be very happy (including the sum that appears in the last line of Alg 1). Similarly, (13) and the paragraph above is not clear. As I understand, an invertible mapping $T$ is utilized to switch between processes with constant and state dependent diffusions. Yet, I'm not able to immediately see (13). It would be much better to clearly define all the terms/processes and show where the (state) independence comes from.

**Summary Of The Paper:**

The paper presents a new approach to compute the density of a given observation sequence under arbitrary drift and diffusion functions. The method relies on Girsanov's theorem, which requires evaluating the density of the observations under Wiener process. The authors further propose an inversion trick to handle state dependent diffusion functions. The experiments show that the method indeed learns accurate drift and diffusion functions much faster than standard integration-based approaches and has much smaller gradient variance

**Summary Of The Review:**

The proposed approach concerns an interesting topic and provides excellent improvements over existing approaches. However, I'm not able to mathematically verify Theorem 1 and the derivation in Section 2.2. This is why I preliminarily vote for a reject but would be very happy to re-consider my score if additional details are provided by the authors and discussions with other reviewers.

---

> ### Author Response · Authors · 2021-11-22
> **Reply to Reviewer k19Y**
>
> Thank you for your feedback. Regarding the dimensionality, as far as we are aware, there are no alternative algorithms which are feasible in higher dimensions than what we have discussed; please see our general response to all reviewers.
>
> Admittedly there are multiple notions of prior/posterior involved here and it can be a difficult point to represent clearly. We have a space for the functions $f,g$ which define the SDE, the space of sample-paths $X$ for a given SDE, discrete time evaluations $X(t_n)$ of a given sample path, and then the observations $x_n$ which arise at the corresponding times, dependent on the value of the path. Each of these spaces has a corresponding prior and posterior distribution.
>
> In line 48, we are specifically referring to the distribution $p(X|f,g)$ of sample-paths for a fixed $f$ and $g$, without conditioning the path distribution on the observations. The corresponding posterior distribution here is the distribution $p(X|f,g,x_{1:N})$ of SDE sample-paths, for the same $f$ and $g$, which are restricted to those passing through the observations. SDE Integrators sample from the prior $p(X|f,g)$, while our method uses importance sampling to estimate expectations over the posterior $p(X|f,g,x_{1:N})$.
>
> There are further notions of prior/posterior which are not discussed in line 48.
> Given a prior $p(f)$ over $f$, there is also an associated posterior distribution $p(f|x_{1:N})$, which we would call the learned SDE. This gives rise to another distribution over paths $\int p(X|f,g)p(df,dg|x_{1:N})$, which do not necessarily pass through the observations (independent realisation of $W_t$), as well as a distribution, $\int p(X|f,g,x_{1:N})p(df,dg|x_{1:N})$, of paths which do pass through the observations (the same realization of $W_t$ as the observations), which may also be called a prior/posterior pair. The prior/posterior over the vector field $f$ is what is modeled as a Gaussian process in our GP experiment.
>
> ## Theorem 1
> Likely the easiest way to see how the exponential arises in theorem 1 is to consider the Euler discretized form of the Ito SDE. The continuous limit of this discretization recovers the Ito SDE.
> $$
> \Delta x_{n+1} = f(x_n)\Delta t + g(x_n) \mathcal{N}(0, \Delta t)
> $$
> $$
> \Delta x_{n+1} \sim \mathcal{N}(f(x_n)\Delta t, g^2(x_n)\Delta t)
> $$
> One can expand the the probability distribution for this discretization (in one dimension here for simplicity, but the calculation is the same for any number of dimensions)
> $$
> p_{X}(\\{x_n\\}\_n) = \frac{1}{\prod_n g(x_n)\sqrt{2\pi\Delta t}} \exp\left(
> -\frac{1}{2}\sum_n \left(
>   \frac{\Delta x_{n+1} - f(x_n)\Delta t}{g(x_n)\sqrt{\Delta t}}
>   \right)^2
> \right)
> $$
> $$
> = \frac{1}{\prod_n g(x_n)\sqrt{2\pi\Delta t}} \exp\left(
> -\frac{1}{2}\sum_n \left(
>   \frac{\Delta x_{n+1}}{g(x_n)\sqrt{\Delta t}}
>   \right)^2
> \right)
> \exp\left(
> \sum_n \frac{f(x_n)}{g^2(x_n)} \Delta x_{n+1}
> -\frac{1}{2} \sum_n \frac{f^2(x_n)}{g^2(x_n)} \Delta t
> \right)
> $$
> Taking an expectation for an arbitrary functional $F[x]$ in the continuum limit we get
> $$
> E_X\left[F[X]\right] = E_Y\left[F[Y] \exp(S[Y]) \right],
> $$
> where $Y$ is the process without drift, and $S$ is the continuum limit of the term in the rightmost exponential above
> $$
> S[X] = \lim \sum_n \frac{f(X_n)}{g^2(X_n)} \Delta X_{n+1}
> -\frac{1}{2} \sum_n \frac{f^2(X_n)}{g^2(X_n)} \Delta t
> $$
> $$
> = \int \frac{f(X)}{g^2(X)} dX
> -\frac{1}{2} \int \frac{f^2(X)}{g^2(X)} dt.
> $$
> Lastly, the conditional expectation arises from an application of Bayes theorem inside the expectation with the conditional distribution $p(x_t | X) = \delta(x_t - X(t))$
> $$
> p_X(x_t) = E_X[\delta(x_t - X(t))] = E_Y\left[\delta(x_t - Y(t)) e^{S[Y]}\right]
> $$
> Conditional expectation being defined as
> $$
> \underbrace{E_Y[F[Y] \,
> \overbrace{\delta(x_t - Y(t))}^{\mathrm{likelihood}}
> ]}\_{\mathrm{prior\, expectation}} =
> \overbrace{p_Y(x_t)}^{\mathrm{evidence}}
> \underbrace{E_Y[F[Y] \,|\, Y(t) = x_t]}\_{\mathrm{posterior\, expectation}}.
> $$
> The final result is
> $$
> p_X(x_t) = E_Y\left[\delta(x_t - Y(t)) e^{S[Y]}\right]
> = p_Y(x_t) E_Y\left[e^{S[Y]} \,\middle|\, Y(t) = x_t\right].
> $$
> The sum in the last line of Alg 1 is the sample mean of the stochastic estimates of the exponential which converges to expectation in equation 9 (up to discretization error) for large $K$.

---

### Official Review · Reviewer_68Rx · 2021-11-03

**Correctness:** 3
**Technical Novelty And Significance:** 3
**Empirical Novelty And Significance:** 1
**Recommendation:** 5
**Confidence:** 3

**Main Review:**

While the proposed method improves the speed and efficiency of SDE learning, the presentation is not strong enough. Below are my specific comments.

* The scope of the work is limited to state-independent or transformable to state-independent diffusion. This is not properly reflected in the title and abstract of the paper and should be inferred by the reader throughout the paper.
* Methodological novelty is limited. The paper is heavily based on a simple application of theorem 1.
* Comparisons are done only on the toy examples, if the authors believe that the proposed method suffices for most application then results on higher dimensions, various simulations, and real world data should be presented. Provided that the technical novelty is limited, empirical validation is necessary for the publication.
* Comparisons against the adjoint method is not completely fair, since it solves a way more general problem (state-dependent diffusion). Are there methods tailored to state-independent diffusion to be used for comparison?
* Why gradient variance results is shown on a separate dataset than learning curves? Please include more than one example for the learning curves experiment and include standard deviation across multiple experiments. Also, can you show this as a function of dimensions, and when adding or removing observation noise?




Here are some minor comments:

* Introduction is very short and many applications of SDE are not mentioned.
* Line 25: "In many physical applications ..." please provide examples and include citations.
* Line 27: "In more complex processes ..."  please provide examples and include citations.
* Line 46: "Models that implicitly describe f as a Gaussian process"  please provide examples and include citations.
* Line 56: "Can marginalize out the observation noise ..." please provide examples of this, how do you marginalize out the observation noise?
* Theorem 1: please spell out the conditions of Girsanov's theorem explicitly and what exactly should hold for function f and sigma.
* Instances of informal language can be found throughout the text such as "wandering around?" or "if need be".
* Line 184: "It may be argued ..." please provide evidence for this. Can you support the argument "almost all real world problems ..."  by providing examples from the literature?
* Some of the text in the supplementary must be brought to the main. In the current form, it's not clear how exactly the experiments are done. I had a grasp on the experiments section after reading the supplementary.
* Variational GP experiment is very unclear, what is the generative model? What is data (include equations)? What is the inference procedure?

**Summary Of The Paper:**

The paper proposes an importance sampling method for the estimation of probabilities in SDE models, applicable to SDEs with state-independent (or transformable to state-independent) diffusion. By taking the advantage of Girsanov's theorem, the probability density of the observations is rewritten as a weighted probability with respect to a base Brownian motion process. Sampling results are presented qualitatively and compared to integration based sampling. Results also suggest that the proposed method is faster than integration based methods and the gradients have lower variances. In addition, a variational GP examples is demonstrated showing that the method can provide reasonable uncertainty estimation.

**Summary Of The Review:**

In summary, the methods presented can be of potential interest to the community, however the presentation is weak, scope and novelty is limited, and experiments are rather minimal.

---

> ### Author Response · Authors · 2021-11-22
> **Reply to Reviewer 68Rx**
>
> Thank you for your feedback on our submission. We have attempted to address as many concerns as possible and have incorporated many of the smaller suggestions into the paper already.
>
> The requirements for Girsanov’s theorem, and therefore also required for theorem 1, are fairly mild assumptions. Firstly, the SDE needs to be well defined; the drift and diffusion terms should be such that a solution to the SDE exists. We believe that this is a fair assumption to make considering the context. This can be guaranteed with e.g. Lipschitz conditions; however those details are not specifically required for the theorem, only for the existence of the SDE itself. Lastly, the Ito integral $S$ should exist and be almost surely finite. Again we believe that this assumption is self-evident.
>
> While we hesitate to make a claim in the paper without a formal proof, we believe that our approach to state-dependent diffusion using bijective transformations is, at least in theory, flexible enough for all applications. In this approach, the diffusion is given by the product of the Jacobian of the transformation and $\sigma$. The invertibility of the transformation implies that the Jacobian matrix is itself also invertible. We have explicitly assumed that $\sigma\sigma^T$ is invertible; however, it may be time-dependent and include arbitrary permutations etc. The only restriction on the resulting diffusion term is that, due to invertibility, the driving Wiener process should not be lower dimensional than the state. In cases where a lower dimensional Wiener process is required, this can be addressed easily by separating the state into a coupled SDE and ODE, the latter of which is simply a deterministic transformation of the stochastic component of the state. This splitting is commonly used in the description of many SDE based models already, such Black-Scholes, second order Langevin dynamics etc. Given this flexibility, we are unaware of any situation in which our approach is not, at least in theory, applicable.

---

### Author Response · Authors · 2021-11-22
**Reply to Reviewers**

We would firstly like to thank the reviewers for their time and comments on our submission. We will do our best to address all concerns that were raised, and we have updated the paper accordingly.

As a point of clarification, we are proposing an algorithm to estimate probabilities for the purposes of learning SDEs from observations. However, we do not aim to convince the reader that SDEs are good models for any particular application, nor that any particular combination of model and learning algorithm is to be preferred. It is up to the user to decide what model to use for their application and whether likelihood optimization, MCMC algorithms, or variational inference is appropriate for their problem. Plenty of other literature already explores these ideas in greater depth. We are merely providing a tool to calculate probabilities which can be used with the corresponding learning algorithm. The theoretical applicability of the model and learning algorithm is no different when using our algorithm or using the classical integrator based approaches.

From a practical standpoint, however, our  proposed algorithm improves on current published methods in two important and distinct ways.

Firstly, the reformulation in terms of importance sampling allows computational speedups and parallelization in a way which is not possible for integrator based algorithms. The importance weights for a  particular proposal distribution can be derived from (the vanilla version of) Girsanov’s theorem. The speed improvements and parallelizability of the importance sampling approach are algorithmic in nature and are not dependent on the model, data set, or the dimensionality of the problem. In contrast, any algorithm based on SDE integrators will suffer from the same performance hurdles. For example, recent work [1] proposes an SDE integration algorithm which enables various improvements over previous methods. However, despite these improvements, the experiments given in [1] are all on data sets which are 1 or 2 dimensional, and the training times for these experiments on modern GPUs (RTX 2080 and A100) are on the order of days.

The second improvement regards numerical stability. This is enabled by a specific choice of proposal distribution, and is the key difference between our theorem 1 and Girsanov’s theorem.
Girsanov’s theorem provides the Radon--Nikodym derivative between two SDEs. In addition to this, our theorem 1 formulates the importance weights by an expectation conditioned on the observations themselves. The conditional expectation is an application of Bayes theorem and is not a Radon--Nikodym derivative between the two SDEs in the sense that Girsanov’s theorem is (the posterior is not absolutely continuous with respect to the prior). This approach enables us to generate trajectories which pass exactly through the observations (or a proxy thereof). In other words, we guarantee that the generated trajectories are already plausible candidates for the process which gave rise to the data. Unlike the previous point, the numerical stability of this choice of importance distribution does depend on the details of the data set and the dimensionality of the problem.

---

> ### Author Response · Authors · 2021-11-22
> **Purpose of the experiments**
>
> ### Lorenz
> As mentioned above, the speed up and parallelizability of our approach does not depend on the detail of the data set. These properties are purely algorithmic and would apply similarly regardless of the problem. Given this, it is still important to verify that our algorithm does indeed provide a performance improvement and to assess the extent to which it does. This is the main purpose of the Lorenz experiment. While we do not believe that training via maximum likelihood is the best approach to learning SDEs, which are inherently noisy, the error curves do indicate that the model is in fact learning the underlying vector field. In this experiment we show that our approach does provide significant speed ups over integrator based approaches. For fair comparison, the exact same code is used for each algorithm; the training loops and optimization parameters are identical, the only difference is the calculation of the loss function, in one case by importance sampling, the other by integrating out the SDE. Hence, the result would be the same for any integrator which imposes a sequential dependence.
>
> Regarding the parallelization, these experiments were performed on a single GPU (an RTX 3070), and GPU utilization was monitored during the process. The integrator approach maxed out at 25% GPU utilization, while our algorithm consistently maintained 100% GPU utilization, except during MSE evaluation which also used an integrator. In both cases, timings exclude evaluation.
>
> ### Gradient Variance
> Unlike parallelizability, the robustness of our choice of importance distribution does indeed depend on the data and dimension of the problem. For this reason we decided to measure the gradient variance of our algorithm on a real world data set: the Hungarian chickenpox data set from the UCI repository. This is by far the highest dimensional data set we have seen in the literature on learning SDEs using neural networks without using some low dimensional embedding.
>
> We would like to stress that we are not claiming that SDE models are the best choice for this data set. Furthermore, a model trained to completion should theoretically provide the same quality of predictions regardless of whether our algorithm or an integrator is used. The difference is in the ability of the learning algorithm to train the model, which in turn depends on the gradient variance, especially at the initial stages of learning.
>
> Much of the literature focusing on the adjoint approach to gradient estimation discusses the gradient error of their respective approaches. This is due to the fact that integrating the SDE backwards in time may not result in the same trajectory, even when the Brownian motion is correctly reconstructed. In these cases, the estimated gradients are usually compared to the analytical gradients when available, or else calculated by directly backpropagating through the SDE solver; see [2]. This gradient error is completely different to the gradient variance experienced during training.
>
> In our experiment we measure the variance of the gradients, per parameter, calculated by directly backpropagating through an SDE solver over many sample paths. Therefore the gradient variance we are calculating is the inherent variance induced by the simple fact that the paths themselves are stochastic. If the variances could be computed analytically, they would be of the same magnitude, and no integration algorithm could reduce them. Our approach on the other hand avoids this high variance by generating trajectories which pass through the observations, and calculating a simple quadratic $S$. The variance of gradients in our algorithm are simply proportional to (the logsumexp of) the typical square magnitude of the vector field $f$ at initialization.

---

> > ### Author Response · Authors · 2021-11-22
> > **Purpose of experiments cont.**
> >
> > ### Accuracy of Probability Estimates
> > Due to space constraints, we had to move this experiment into the appendix. We used our algorithm to estimate the probability of a set of observations under the assumption of an Ornstein--Uhlenbeck process. This process in our example is independent across dimensions, and so its probability factorizes as such. However, the path-space posterior distribution is highly correlated across dimensions, and so this still poses a challenge for techniques such as importance sampling, which is known to have deteriorating performance in high dimensions. The purpose of this experiment was to verify that importance sampling with our specific choice of importance distribution would be reliable in estimating probabilities despite challenges with the curse of dimensionality.
> > Our results showed that our technique did in fact provide accurate probability estimates, even up to 100 dimensions. Importantly, we did not assume that the data resembled samples from the Ornstein--Uhlenbeck process, as real world observations do not typically resemble trajectories from a neural SDE with randomly initialized weights. **Again, 100 dimensions is significantly larger than any other experiment we have encountered in the literature**.
> >
> > ### Gaussian Process Vector Field
> > Lastly the purpose of our Gaussian process experiment is simply to illustrate that the removal of the sequential dependence of the integrator greatly simplifies and enables the use of Gaussian process based models. We do not wish to imply that the choice of kernel is at all reasonable for the problem, nor that the resulting fit actually exhibits Gaussian errors. We only wish to show that variational Gaussian processes are a computationally viable option for SDE learning when our algorithm is used. In contrast, conditional sampling of a Gaussian process becomes intractable when an SDE integrator is used instead.

---

> > > ### Author Response · Authors · 2021-11-22
> > > **On dimensionality of experiments**
> > >
> > > It has been mentioned by the reviewers that our experiments are fairly small-scale. However, we would like to point out that learning SDEs from observations using neural networks is in and of itself an inherently difficult task in high dimensions. Even in recent work, despite numerous contributions, only include experiments on very low dimensional systems. In contrast our gradient variance experiment was performed on 20 dimensional data without embedding, and our probability estimates were reliable even in 100 dimensions.
> > >
> > > For example [1] introduces various improvements over [2], yet the experiments, which were performed on powerful GPUs, took on the order of days to complete. The dimensionality of observations in each experiment were as follows:
> > >
> > >  - Neural network weights: 1 dimensional
> > >  - Beijing air quality: 2 dimensional
> > >  - Time-dependent Ornstein–Uhlenbeck process: 1 dimensional.
> > >
> > > [2] test their method on various one dimensional toy problems, on the 3 dimensional Lorenz system, and their largest system (the motion capture data set, which has 50 dimensional observations) models the SDE in a latent space of dimension 6.
> > >
> > > [3] and [4] perform experiments on various toy systems of between 1 and 3 dimensions. [4] also uses the motion capture data set, but projects the data down to a 3 dimensional subspace. Finally [5] performs a few experiments on a 1 dimensional dataset and an experiment on the Beijing air quality data set on a 6 dimensional subspace. As of yet, we are unaware of any work which includes experiments in higher dimensional systems than those we have mentioned.
> > >
> > > We do not wish to criticise any of the above mentioned work, or to use them as an excuse. We simply believe that the lack of high dimensional experiments in the literature is a testament to the inherent difficulty of learning SDEs in such contexts.
> > >
> > >
> > > ### References
> > > [1] Kidger, Patrick, et al. "Efficient and Accurate Gradients for Neural SDEs." arXiv preprint arXiv:2105.13493 (2021).
> > >
> > > [2] Xuechen Li, Ting-Kam Leonard Wong, Ricky T. Q. Chen, and David Duvenaud. Scalable gradients for stochastic differential equations. International Conference on Artificial Intelligence and Statistics, 2020.
> > >
> > > [3] Philipp Batz, Andreas Ruttor, and Manfred Opper. Approximate Bayes learning of stochastic differential equations. Phys. Rev. E, 98:022109, Aug 2018. doi: 10.1103/PhysRevE.98.022109.
> > >
> > > [4] C. Yildiz, M. Heinonen, J. Intosalmi, H. Mannerstrom, and H. Lahdesmaki. Learning stochastic differential equations with gaussian processes without gradient matching. In IEEE International Workshop on Machine Learning for Signal Processing. MLSP, 2018.
> > >
> > > [5] Patrick Kidger, James Foster, Xuechen Li, and Terry J Lyons. Neural SDEs as Infinite-Dimensional 336 GANs. In Marina Meila and Tong Zhang (eds.), Proceedings of the 38th International Conference 337 on Machine Learning, volume 139 of Proceedings of Machine Learning Research, pp. 5453–5463. 338 PMLR, 18–24 Jul 2021.

---

### Decision · Program_Chairs · 2022-01-20

**Decision:**

Accept (Poster)

**Comment:**

This paper proposes an importance-sampling estimator for probabilities of observations of SDEs. The proposed approach has several advantages over conventional methods: it does not require an SDE solver, it has lower gradient variance, and shows nice results with a Gaussian process representation of the function. Reviewers were somewhat split on this paper, with some concerns that experiments were limited. On balance, however, the paper makes several nice contributions, the experiments are in line with related works, and the authors did a good job of clarifying Theorem 1 in the rebuttal. We note that Reviewer K19Y changed their opinion to accept (although they forgot to update the score). Please carefully account for all reviewer comments in the final version.